# MC-PRPA-HLFIA Cascade Detection System for Point-of-Care Testing Pan-Drug-Resistant Genes in Urinary Tract Infection Samples

**DOI:** 10.3390/ijms24076784

**Published:** 2023-04-05

**Authors:** Jin Tao, Dejun Liu, Jincheng Xiong, Wenchong Shan, Leina Dou, Weishuai Zhai, Yang Wang, Jianzhong Shen, Kai Wen

**Affiliations:** Beijing Advanced Innovation Center for Food Nutrition and Human Health, College of Veterinary Medicine, China Agricultural University, Beijing 100083, Chinaliudejun@cau.edu.cn (D.L.); xiongjincheng94@163.com (J.X.); shanwenchong@163.com (W.S.); b20193050410@cau.edu.cn (L.D.); zws18363976198@163.com (W.Z.); wangyang@cau.edu.cn (Y.W.)

**Keywords:** extensively drug-resistant genes, Chelex-100, recombinase polymerase amplification, lateral flow immunoassay, urinary tract infection

## Abstract

Recently, urinary tract infection (UTI) triggered by bacteria carrying pan-drug-resistant genes, including carbapenem resistance gene *bla*_NDM_ and *bla*_KPC_, colistin resistance gene *mcr-1*, and *tet*(X) for tigecycline resistance, have been reported, posing a serious challenge to the treatment of clinical UTI. Therefore, point-of-care (POC) detection of these genes in UTI samples without the need for pre-culturing is urgently needed. Based on PEG 200-enhanced recombinase polymerase amplification (RPA) and a refined Chelex-100 lysis method with HRP-catalyzed lateral flow immunoassay (LFIA), we developed an MCL-PRPA-HLFIA cascade assay system for detecting these genes in UTI samples. The refined Chelex-100 lysis method extracts target DNA from UTI samples in 20 min without high-speed centrifugation or pre-incubation of urine samples. Following optimization, the cascade detection system achieved an LOD of 10^2^ CFU/mL with satisfactory specificity and could detect these genes in both simulated and actual UTI samples. It takes less than an hour to complete the process without the use of high-speed centrifuges or other specialized equipment, such as PCR amplifiers. The MCL-PRPA-HLFIA cascade assay system provides new ideas for the construction of rapid detection methods for pan-drug-resistant genes in clinical UTI samples and provides the necessary medication guidance for UTI treatment.

## 1. Introduction

Antimicrobial resistance has spread widely due to the extensive usage of antimicrobial agents, which poses a long-term threat to public health. Pan-drug-resistant bacteria persist in many healthcare contexts, yielding a variety of acute and nosocomial infections. Urinary tract infection (UTI) is the most prevalent nosocomial illness, with almost 50% of human beings having at least one infection in their lifetime [1]. Worryingly, broad-spectrum antibiotics have been used to treat clinical infections, which has led to an increase in UTIs caused by pan-drug-resistant bacteria [2]. Among them, bacterial resistance to the last-resort antibiotics (carbapenem, colistin, tigecycline) is increasing, which has a significant impact on their application in the management of medical infections. The carbapenemases NDM and KPC are responsible for the majority of clinical carbapenem-resistant Enterobacteriaceae (CRE) infections. However, due to their varying aztreonam susceptibilities, it is necessary to explicitly differentiate between them before administering the drug to prevent antibiotic misuse [3]. The recently discovered tigecycline resistance gene *tet*(X) variants and the transferrable colistin resistance gene *mcr-1* have an impact on the treatment of clinical CRE infections [4,5]. Furthermore, infection caused by bacteria harboring pan-drug-resistant genes has recently been reported in UTI, where the detection rate of *mcr-1* and *bla*_NDM_/*bla*_KPC_-positive CRE was approximately 11.4% and 16.6%, respectively [6,7,8]. Therefore, it is crucial to quickly identify these pan-drug-resistant genes in urine samples to enable the administration of appropriate antimicrobial medicines for the treatment of UTIs and to ensure the best possible therapeutic outcomes. Currently, urine samples are taken for culture on agar plates, and then PCR is performed on the obtained bacteria to detect pan-drug-resistant genes, which requires about 3–7 days [9]. This could result in inappropriate use of antimicrobial agents during medication administration and delayed clinical prognosis. Hence, a highly sensitive assay is required to reduce the time needed to identify pan-drug-resistant genes in UTI samples for POC testing.

Rapid in situ detection at the POC is made possible by isothermal amplification because it eliminates the need for complex and bulky thermal cyclers, and it allows constant and low-temperature amplification of target genes, such as loop-mediated isothermal amplification (LAMP), strand displacement amplification (SDA), and recombinase polymerase amplification (RPA) [10]. Of these, RPA involves a strand exchange reaction based on recombinase and strand-substituted DNA polymerase, which occurs at 37–43 °C and achieves sensitive nucleic acid amplification within 30 min [11].

To detect pan-drug-resistant genes, however, it is also necessary to extract the target DNA and identify the results of nucleic acid amplification. Current commercial nucleic acid extraction kits take over 1 h to extract DNA from urine samples, requiring bulky high-speed centrifuges, which make rapid detection and analysis challenging. Consequently, a method for the rapid extraction of DNA from urine samples appropriate for on-site use is warranted. Previously, we used a lateral flow immunoassay (LFIA), a strip-based platform using antigen–antibody specific binding, to allow rapid in-field detection of RPA amplification products [12].

Therefore, for POC testing the pan-drug-resistant gene in UTI samples, an MCL-PRPA-HLFIA cascade detection system was designed based on previously developed PEG 200-enhanced RPA, along with a refined Chelex-100 lysis technique and HRP-catalyzed LFIA. Within an hour, the system performs POC detection of genes with high sensitivity and specificity, providing appropriate dosing guidance for the treatment of clinical UTI.

## 2. Results

### 2.1. Principles

#### 2.1.1. Nfo-RPA

The reverse primers and probes were, respectively, biotin and digoxin/carboxytetramethylrhodamine (TAMRA)/FITC/Cy3 tagged. (Figure 1A). The unlabeled forward primer first formed an amplicon with the labeled reverse primer, then hybridized with the probe to produce an amplicon consisting of the probe and the reverse primer.

#### 2.1.2. LFIA Catalyzed by HRP

The HRP (horseradish peroxidase)-catalyzed LFIA was used to identify the double-stranded DNA amplicons produced by PEG 200-enhanced nfo-RPA (Figure 1B). When the target gene was present in the amplicon, the digoxin/carboxytetramethylrhodamine (TAMRA)/FITC/Cy3 moiety of the amplicon was captured by the corresponding antibody on the test line, resulting in a colorimetric signal. Conversely, no colorimetric signal appeared on the test line. Moreover, the complex of AuNPs and anti-biotin (HRP conjugate) was captured by IgG on the control line, resulting in a colorimetric signal on the line, demonstrating that the test strip operated properly. Furthermore, HRP could catalyze the reaction between AEC (3,3′,5,5′-3-Amino-9-Ethylcarbazole, the substrate of HRP) and H_2_O_2_, the colored products attached to the surface of AuNPs, increasing the test line’s and control line’s colorimetric signal [13].

### 2.2. Within 20 min, a Modified Chelex-100 Lysis Technique Could Recover DNA from Urine Containing Uncultured Bacteria 

First, we found that employing the modified Chelex-100 lysis technique to extract DNA from a 500 μL sample of synthetic urine containing bacteria resulted in a greater concentration of DNA without wasting the sample (Appendix A). After validation, bacterial DNA could be extracted from urine samples containing 10^7^–10^0^ CFU/mL of *mcr-1*-positive bacteria using this method, and the concentration of double-stranded DNA decreased from 1.03 to 0.14, with OD_260_/OD_280_ and OD_260_/OD_230_ at around 1.65 and 1.61, respectively (Figure 2). Since bacterial DNA could not be extracted from uncultured urine samples using the commercial kit (QIAamp^®^ BiOstic^®^ Bacteremia DNA Kit, Qiagen Co., Ltd., Dusseldorf, Germany), samples were incubated overnight at 37 °C to compare the extraction performance of the two procedures. Our method could extract bacterial DNA from cultured urine samples containing 10^7^–10^0^ CFU/mL bacteria, while commercial kits could not extract bacterial DNA from the samples containing 10^1^ and 10^0^ CFU/mL bacteria (Figure 3). Additionally, our technique made it possible to extract bacterial DNA from 95 urine samples containing 10^2^ CFU/mL bacteria prepared by clinically collected UTI samples, with the extracted DNA concentration in the range of 1.02–1.81 ng/μL, in less than 20 min, and without the use of a bulky high-speed centrifuge (Appendix A).

### 2.3. Successful Fabrication of AuNPs and HRP-AuNPs-Antibody Conjugates 

Transmission electron microscopy (TEM) revealed that the AuNPs were round spheres with a relatively regular shape and an average diameter of 32 nm (Appendix A). The highest wavelength of unlabeled AuNP absorption was 528 nm, according to the ultraviolet-visible spectrum, in contrast to the maximum absorption wavelength red-shifted to 534 nm after labeling with anti-biotin (HRP conjugate, Appendix A). The zeta potential of AuNPs was −30.93 mV, while it changed to −21.53 mV after antibody adsorption (Appendix A). The above results demonstrated that AuNPs were successfully synthesized, and the HRP-AuNPs-antibody conjugate was conducted satisfactorily.

### 2.4. Optimization of the Detection Conditions

#### 2.4.1. PEG 200 Enhanced RPA

Due to the RPA reaction circumstances used in this study, for the remaining above-mentioned pan-drug-resistant genes, except for *tet*(X) determined in our previous study, only the conditions for the detection of *tet*(X) were determined [12]. According to the results, the final fluorescence intensity was greatest at 41 °C after 15 min (Appendix A). Therefore, this set of reaction conditions was determined to be the optimal set of conditions for the RPA amplification of *tet*(X).

#### 2.4.2. HRP-Catalyzed LFIA

After determining the RPA reaction conditions for these genes, the amount of anti-biotin (HRP conjugate) coated during conjugate preparation and the volume of HRP-AuNPs-antibody conjugate were both optimized. A quantity of 5 μL of template DNA was added at a concentration of approximately 15 ng/μL during the detection of the pan-drug-resistant genes. Taking the detection of *mcr-1* as an example, the results revealed that the addition of 6 µg of anti-biotin was appropriate for antibody conjugation, which caused the test line to have the highest colorimetric signal. In addition, the usage of the complex of AuNPs and anti-biotin (HRP conjugate) was 3 µL (Appendix A).

### 2.5. The Cascade System’s Sensitivity for Detecting Pan-Drug-Resistant Genes in Urine Samples Reached 10^2^ CFU/mL

The detection system’s sensitivity was determined by detecting the pan-drug-resistant genes (*mcr-1*, *bla*_NDM_, *bla*_KPC_, *tet*(X)) in urine samples containing different concentrations (10^7^–10^0^ CFU/mL) of pan-drug-resistant bacteria under optimized conditions (Figure 4). As the concentration of the target-gene-positive bacteria in the urine samples decreased, the test line’s colorimetric signal was gradually lost, and no colorimetric signal appeared in the test line of the negative samples (urine samples containing no pan-drug-resistant bacteria). We defined the minimum concentration of pan-drug-resistant bacteria in the urine sample when the test line was invisible as the visual limit of detection (vLOD) for this cascade detection system. Thus, the vLOD of the designed detection system was 10^2^ CFU/mL for *mcr-1*, *bla*_NDM_, and *bla*_KPC_ in the urine samples. The prototype of *tet*(X) and its variant *tet*(X2) are generally present in usually unavailable anaerobic unculturable bacteria, so they were not tested, and the vLOD for the remaining variants of *tet*(X) was 10^2^ CFU/mL.

### 2.6. The Cascade Detection System Has Favorable Specificity

To validate the specificity of the MCL-PRPA-HLFIA cascade assay system to distinguish the target pan-drug-resistant genes from other interfering genes, we tested the *mcr-1*, *bla*_NDM_, *bla*_KPC_, *tet*(X), *bla*_OXA-1_, and ATCC 25922 in urine samples (Figure 5). The above-mentioned gene-positive bacteria and ATCC 25922 were spiked into the synthetic urine sample at a concentration of 10^7^ CFU/mL. Only when the target gene was found in the sample was the colorimetric signal displayed simultaneously in the test and control lines. In the other samples, this test line’s colorimetric signal was hardly detectable, indicating that the assay’s overall specificity was good.

### 2.7. Evaluation of Cascade Detection Method Sensitivity and Specificity in Identifying the Pan-Drug-Resistant Genes in Simulated and Real UTI Samples

The MCL-PRPA-HLFIA cascade detection system was first applied to 95 mock UTI samples to ensure its viability in actual applications. Direct access to the detection system’s sensitivity and specificity was possible by testing these samples since the pan-drug-resistant-positive bacteria included in the manufacture of the simulated UTI samples were identified by second-generation sequencing of the corresponding genes. For simulated UTI samples, the constructed assay’s sensitivity for *mcr-1*, *bla*_NDM_, *bla*_KPC_, *tet*(X) was 96.0% (95%CI: 79.6–99.9%), 95.0% (95%CI: 75.1–99.9%), 95.5% (95%CI: 77.2–99.9%), 95.8% (95%CI: 78.9–99.9%), respectively, while the specificity was over 97% for the simulated UTI sample (Table 1). Subsequently, we detected the pan-drug-resistant genes in clinically collected UTI samples using the general clinical method (GCM): directly culturing UTI samples on solid media, then extracting bacterial DNA with commercial kits, and finally identifying these genes using PCR. Additionally, we tested these samples simultaneously using the detection system we constructed. For actual UTI samples, the sensitivity for *mcr-1* and *bla*_KPC_ was 100.0%, and the specificity for *mcr-1*, *bla*_NDM_, *bla*_KPC_, and *tet*(X) was 100.0% (95%CI: 88.8–100.0%), 100.0% (95%CI: 89.1–100.0%), 96.8% (95%CI: 83.3–99.9%), and 96.9% (95%CI: 83.8–99.9%), respectively (Table 2).

## 3. Discussion

The prevalence of UTI caused by bacteria harboring pan-drug-resistant genes, such as *mcr-1*, *bla*_NDM_, *bla*_KPC_, and *tet*(X), has become increasingly significant [7,14,15]. In UTI samples, clinical testing for a pan-drug-resistant gene involves bacterial culture, which takes three days to obtain results, or PCR-based methods, which require specialized and complex equipment. In both cases, testing cannot be performed at the POC, delaying treatment, and resulting in inappropriate antibiotic use [16]. To detect the pan-drug-resistant gene, we constructed the MCL-PRPA-HLFIA cascade assay using PEG 200-enhanced RPA along with modified Chelex-100 lysis and HRP-catalyzed LFIA. This approach could detect these genes in UTI samples within 1 h without the need for complex instruments, such as high-speed centrifuges and PCR machines.

With a bacterial content of 10^1^ CFU/mL, our refined Chelex-100 lysis technique was able to extract bacterial DNA from uncultured UTI samples without using a high-speed centrifuge. With commercial kits, nucleic acid could only be extracted from samples that have been pre-cultured and contain at least 10^2^ CFU/mL of bacteria, which is perhaps due to the need to enrich the samples by centrifugation before the extraction process. Moreover, the modified method could accomplish the same task in a shorter time (within 20 min) than microfluidic-based devices for UTI sample pre-processing [17]. Thus, the constructed modified Chelex-100 lysis method enables rapid on-site extraction of bacterial DNA from UTI samples.

A cascade detection system based on RPA was constructed for this study, which offers greater detection sensitivity than PCR-based tests. For the detection of pan-drug-resistant genes in UTI samples, the multiplex PCR-based assay had an LOD of 10^3^ CFU/mL, which was inferior to the cascade assay [18]. The latest nanopore sequencing technology could also be used for the detection of these genes in UTI samples, with an accuracy comparable to that of commonly used clinical methods; however, this assay still requires pre-culture of UTI samples, which is time-consuming [19].

Due to the ability to perform parallel detection, DNA microarrays can also detect multiple pan-drug-resistant genes and virulence genes in bacteria with good detection accuracy [20]. The LOD of our MCL-PRPA-HLFIA cascade detection system for *bla*_KPC_ in urine samples was 10^2^ CFU/mL, which was slightly better than that (360 CFU/mL) of the DNA microarray system [21]. In addition, a commercial kit was used for preprocessing the samples in this study, which required a long period and high-speed centrifuge assistance. Since a bacterial count of more than 10^5^ CFU/mL is typically indicative of UTI, a minimal bacterial count of 10^2^ CFU/mL needed in our cascade assay system would be a more accurate standard for symptomatic patients, which further satisfy the requirement for detecting clinical UTI. Furthermore, our system’s specificity and sensitivity for detecting carbapenemases in actual urine samples were 100% and 96.8%, respectively, similar to MALDI-TOF, which has 100% sensitivity and specificity [22].

The MCL-PRPA-HLFIA cascade detection system could detect pan-drug-resistant genes in UTI samples within 1 h without pre-incubation and use of professional instruments other than metal baths, and offers favorable clinical application prospects and provides guidance for the treatment of clinical UTIs to prevent the development of antimicrobial resistance resulting from improper use of antibacterial agents.

Although the system could aid in the rapid clinical selection of available drugs without sample preculturing, it is not available to cover all pan-drug-resistant genes, and we have only selected important genes in specific bacterial species in the development of the method. Moreover, the pre-treatment, nucleic acid amplification and product detection parts of the cascade assay system could only be performed independently of each other and not in the same system, thus making the whole assay process somewhat complicated. Therefore, it would be more promising if the component processes could be integrated with microfluidics to reduce the independent preprocessing, amplification, and detection steps.

## 4. Materials and Methods

### 4.1. Materials and Chemicals

Chelex-100, Triton X-100, chloroauric acid tetrahydrate (HAuCl_4_·3H_2_O), trisodium citrate, sodium chloride (NaCl), polyethylene glycol (PEG) 200, PBS and bovine serum albumin (BSA) were purchased from Sinopharm Chemical Reagent Beijing Co., Ltd. (Beijing, China). The zymolyte for the HRP catalyzed reaction was received from Keyuezhongkai Biotechnology Co., Ltd. (Beijing, China). The synthetic urine was provided by the Solarbio Science and Technology Co., Ltd. (Beijing, China). TE buffer and nuclease-free water were received from Magen Biotechnology Co., Ltd. (Guangzhou, China). The anti-biotin, anti-digoxin, anti-rhodamine, anti-FITC, and anti-Cy3 were purchased from Abcam Inc. (Cambridge, MA, USA). The goat anti-rabbit IgG was purchased from Jackson Immuno- Research Co., Ltd. (West Grove, NJ, USA).

### 4.2. Construction of Standard Bacterial Strains

The detailed preparation procedure is displayed in the Appendix A.

### 4.3. Preparation of Simulated UTI Samples

Thirty-two actual UTI samples were collected in 2022 by the General Hospital of the People’s Liberation Army. A quantity of 4 mL of each clinically collected UTI sample was aspirated and mixed, then filtered using a 0.22 µm filter membrane to make a mock urine solution. To make 95 simulated UTI samples, 25 *mcr-1*-positive, 20 *bla*_NDM_-positive, 22 *bla*_KPC_-positive, 24 *tet*(X)-positive, 1 *mcr-3*-positive, 1 *bla*_OXA-1_-positive, 1 ATCC 13883 and 1 ATCC 25922 strain were introduced to mock urine at a concentration of 10^2^ CFU/mL, respectively.

### 4.4. Bacterial DNA Extraction from the Bacteria-Containing Urine Sample

First, a 100 µL bacterial solution of 10^8^–10^1^ CFU/mL was added to 900 µL of synthetic urine to prepare urine samples containing 10^7^–10^0^ CFU/mL of bacteria.

Previously, we constructed a procedure for extracting DNA from bacteria using the modified Chelex-100 lysis method [12,23]. In this study, we attempted to use the approach to extract bacterial DNA directly from urine samples. The specific workflow is as follows:To prepare the Chelex-100 lysis solution, combine 2.5 g Chelex-100, 50 mL TE buffer, and 500 µL TritonX-100 in a container and thoroughly mix.Add 200 µL of Chelex-100 lysis solution to the prepared bacterial-containing urine sample and mix completely.Heat the mixture at 100 °C for 10 min.A 1 mL syringe is used to aspirate the mixture, which is then filtered by a 0.45 µm filter membrane.

Analyzing the extracted DNA quality (OD_260_/OD_280_, OD_260_/OD_230_) was carried out with a UV-Vis spectrophotometer (Thermo Fisher Scientific, Waltham, MA, USA). Using Qubit (Thermo Fisher Scientific, USA), the concentration of double-stranded DNA was quantified.

### 4.5. PEG 200 Enhanced RPA

#### 4.5.1. Primers and Probes Design for RPA

The RPA primers and probes for the target genes were designed by PrimedRPA (Appendix A) [24].

#### 4.5.2. Enhanced RPA by PEG 200

The procedure of PEG 200-enhanced exo-RPA was performed based on our previous study; detailed information is displayed in the Appendix A [12].

### 4.6. HRP-Catalyzed Lateral Flow Immunoassay

#### 4.6.1. Synthesis of HRP-AuNPs-Antibody Conjugate

A previously described procedure was used to generate gold nanoparticles (AuNPs) [25]. The fabrication of the HRP-AuNPs-antibody conjugate was performed based on previous studies with slight modifications [26]. A quantity of 40 μL of anti-biotin (HRP conjugate) was added to 1 mL of AuNPs solution. Then, 50 μL of a 20% BSA solution was added after standing for 1 h, and the mixture was then incubated for an additional 1 h. After that, the precipitate was kept by centrifuging at 4 °C for 10 min at 11,000 rpm. Finally, 100 μL of resuspension buffer was added to the precipitate for resuspending at room temperature, resulting in the ultimate detection probe for LFIA.

#### 4.6.2. Manufacture of the Strip

The strip was made up of four components, including a sample pad, NC membrane, absorbent pad, and PVC backing card. Anti-digoxin antibody (or anti-rhodamine antibody, or anti-FITC antibody, or anti-Cy3 antibody) solution (1.0 mg/mL) and goat anti-rabbit IgG solution (1.0 mg/mL) were applied on the NC membrane at a rate of 1 μL/cm to construct the test and control zone using the dispensing system purchased from Kinbio Tech. Co., Ltd. (Shanghai, China). The NC membrane was then dried for a whole night at room temperature. Afterward, the above-mentioned components were joined on the PVC backing card. Individual strips of LFIA of 3.0 mm thickness were obtained and kept at 4 °C.

### 4.7. Cascade System for Detection of Pan-Drug-Resistant Gene in Uncultured Urine Samples

DNA extraction was carried out using a refined Chelex-100 lysis method. After that, PEG 200-enhanced nfo-RPA was implemented using the nfo-RPA lyophilization kit provided by Hangzhou Zhongce Biotechnology Co., Ltd., Hangzhou, China. The reaction system comprised 12.5 μL rehydration buffer, 2 μL primer-F, 2 μL primer-R-nfo, 0.6 μL probe-nfo, 0.5 μL PEG 200, 5 μL DNA template, 24.9 μL nuclease-free water, and 2.5 μL MgOAc. The mixed reaction system was then heated at 41 °C/43 °C for 15 min. Then, the prepared detection probe, 150 μL of PBS, and 25 μL of amplification product were mixed. Subsequently, LFIA strips were vertically inserted into the mixed solution. Ten minutes later, 30 μL of the substrate solution was added dropwise on the strip. Negative results showed a colorimetric signal on the control line exclusively under visual evaluation within 5 min, while positive results showed a signal on both the test and control zones. The entire detection process could be accomplished in less than one hour and required no bulky instruments.

### 4.8. Statistical Analysis

GraphPad Prism 8 was used to plot the graphs and conduct the linear regression analysis. The VassarStats online statistical program was used to determine the 95% confidence interval (95%CI).

## 5. Conclusions

To summarize, in view of the limitations of conventional methods for pan-drug-resistant gene detection in UTI samples, we constructed a MCL-PRPA-HLFIA cascade detection system by combining PEG 200-enhanced RPA with a refined Chelex-100 lysis technique and HRP-catalyzed LFIA. The LOD for detecting the target gene in UTI samples was 10^2^ CFU/mL, with favorable specificity and satisfactory results in both simulated and actual UTI samples. Considering its sensitivity, specificity, simplicity, and rapidity, this work could contribute to the development of rapid detection strategies for pan-drug-resistant genes in clinical UTI samples. A slight modification to the detection system will enable it to detect more pan-drug resistance genes, with great promise for use in clinical and experimental screening and monitoring.

## Figures and Tables

**Figure 1 ijms-24-06784-f001:**
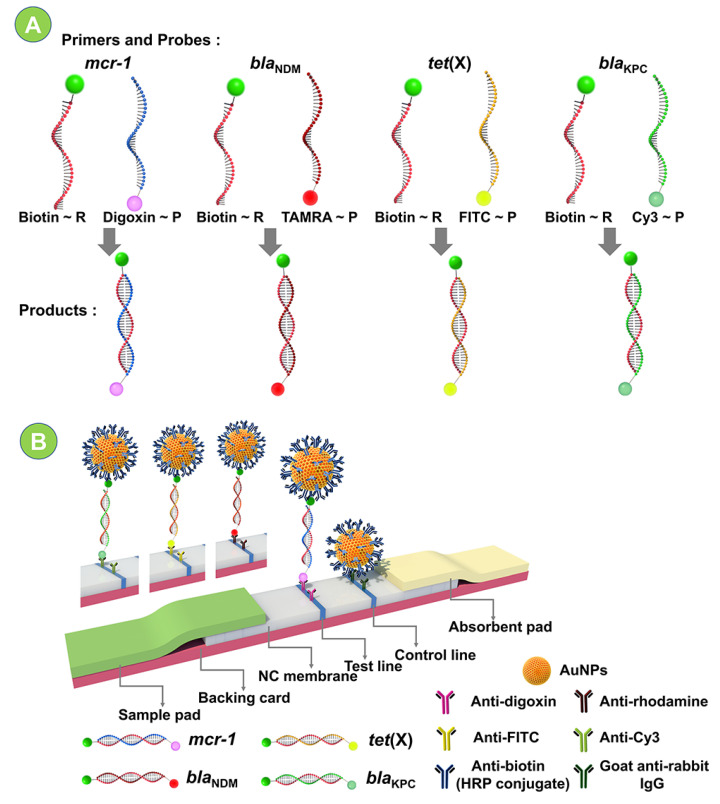
Scheme of the cascade detection system. (**A**) Schematic illustration of RPA, (**B**) HRP-catalyzed LFIA.

**Figure 2 ijms-24-06784-f002:**
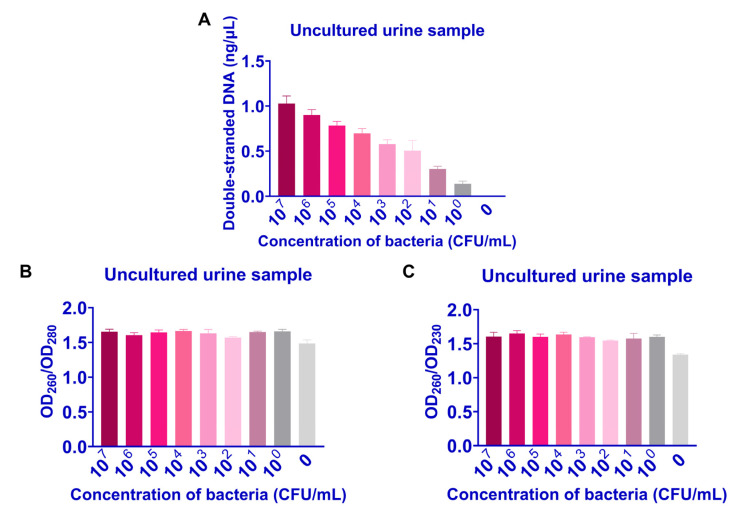
Feasibility of bacterial DNA extraction from uncultured bacterial-containing urine samples by refined Chelex-100 lysis method. (**A**) DNA concentration, (**B**) OD_260_/OD_280_ value, (**C**) OD_260_/OD_230_ value.

**Figure 3 ijms-24-06784-f003:**
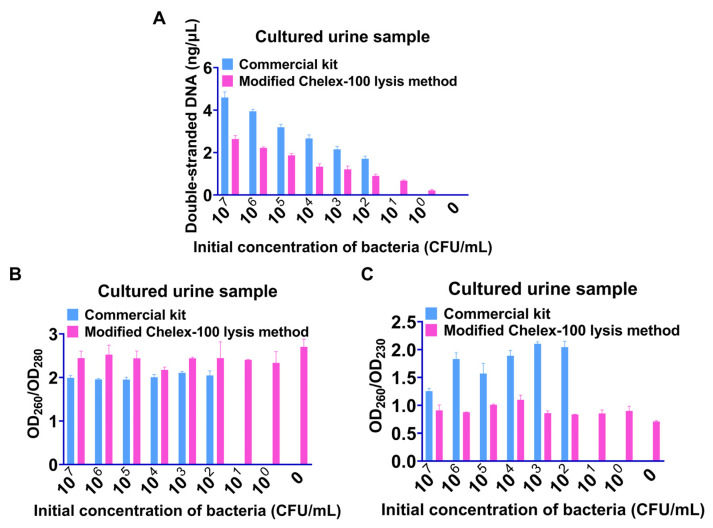
Comparison of the refined Chelex-100 lysis method with commercial kits applied in cultured bacterial-containing urine. (**A**) concentration of DNA, (**B**) OD_260_/OD_280_ value, (**C**) OD_260_/OD_230_ value.

**Figure 4 ijms-24-06784-f004:**
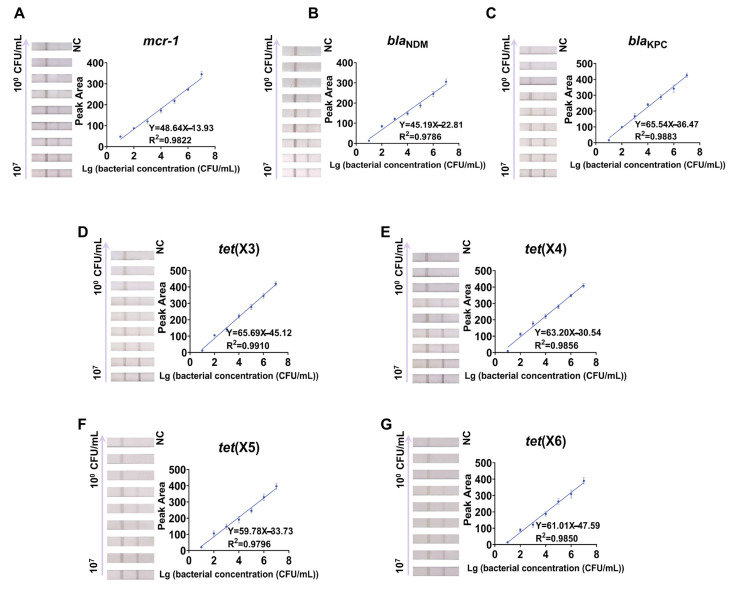
Sensitivity for detecting pan-drug-resistant genes in urine samples. Linear regression and actual images. (**A**) *mcr-1*, (**B**) *bla*_NDM_, (**C**) *bla*_KPC,_ (**D**) *tet*(X3), (**E**) *tet*(X4), (**F**) *tet*(X5), (**G**) *tet*(X6). NC, negative control.

**Figure 5 ijms-24-06784-f005:**
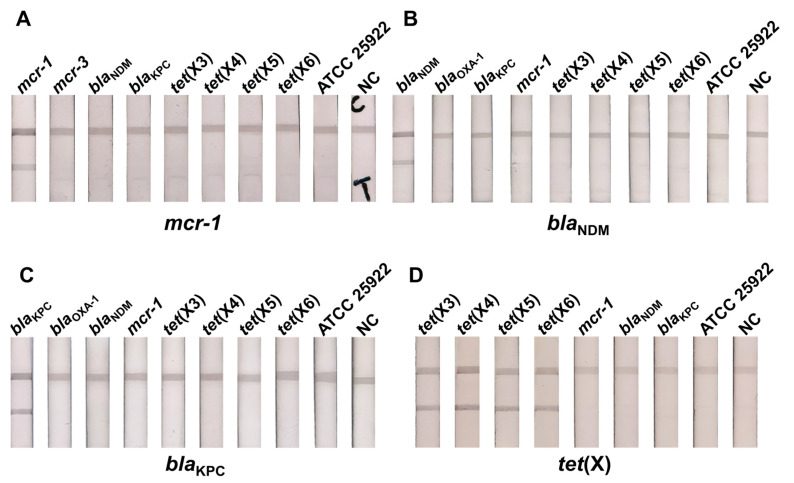
Specificity analysis for pan-drug-resistant genes testing in urine samples. (**A**) *mcr-1*, (**B**) *bla*_NDM_, (**C**) *bla*_KPC,_ (**D**) *tet*(X). NC, negative control.

**Table 1 ijms-24-06784-t001:** Detection system sensitivity and specificity for detecting pan-drug-resistant genes in the simulated UTI sample.

Pan-Drug Resistant Genes	Actually (+), MCL-PRPA-HLFIA (+)	Actually (+), MCL-PRPA-HLFIA (−)	Actually (−), MCL-PRPA-HLFIA (+)	Actually (−), MCL-PRPA-HLFIA (−)	Sensitivity (%)	Specificity (%)
*mcr-1*	24	1	1	69	96.0 (24/24 + 1)	98.6 (69/69 + 1)
*bla* _NDM_	19	1	2	73	95.0 (19/19 + 1)	97.3 (73/73 + 2)
*bla* _KPC_	21	1	2	71	95.5 (21/21 + 1)	97.3 (71/71 + 2)
*tet*(X)	23	1	2	69	95.8 (23/23 + 1)	97.2 (69/69 + 2)

**Table 2 ijms-24-06784-t002:** Detection system sensitivity and specificity for detecting pan-drug-resistant genes in the actual UTI sample.

Pan-Drug Resistant Genes	GCM (+), MCL-PRPA-HLFIA (+)	GCM (+), MCL-PRPA-HLFIA (−)	GCM (−), MCL-PRPA-HLFIA (+)	GCM (−), MCL-PRPA-HLFIA (−)	Sensitivity (%)	Specificity (%)
*mcr-1*	1	0	0	31	100.0 (1/0 + 1)	100.0 (31/0 + 31)
*bla* _NDM_	0	0	0	32	-	100.0 (32/0 + 32)
*bla* _KPC_	1	0	1	30	100.0 (1/0 + 1)	96.8 (30/1 + 30)
*tet*(X)	0	0	1	31	-	96.9 (31/1 + 31)

GCM: general clinical method.

## Data Availability

All data are contained in the manuscript.

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
