# Peer review of "MC-PRPA-HLFIA Cascade Detection System for Point-of-Care Testing Pan-Drug-Resistant Genes in Urinary Tract Infection Samples"

_ijms, 2023, doi:10.3390/ijms24076784_

Round 1

Reviewer 1 Report

Dear all,

I have read the manuscript thoroughly. I donot have major comments and I accept to publish the current fromof the manuscript.

Author Response

Thank you for your approval of this study and good luck with your career!

Reviewer 2 Report

Title: MC-PRPA-HLFIA cascade detection system for point-of-care testing
pan-drug resistant genes in urinary tract infection samples

Journal: International Journal of Molecular Sciences

Review report: The authors present the design and application of a system for the determination of antibiotic resistance genes, NDM, KPC, mcr-1 and tetX, that confer resistance to carbapenems, colistine and tigeclycline. This resistance genes are reported as important in the determination of pandrogo-resistant bacteria isolated from urinary tract infection (ITU). It is a point-of-care test based on a combination  of rapid lysis, polimerase recombinase gene amplification, and a lateral flow immunoassay.

The manuscript is clear, well structured, properly referenced, easy to read and the topic is relevant.

The authors presented a properly squeme to show the principle of the detection system. The results are presented in appropiated tables and figures and are supported by statistical test and supplementary material.

Specific comments: After reviewing the manuscript I suggest the following minor corrections:

Page 2           Line add the meaning of HRP and AEC

Page 5           line 139 include  references related to previous studies

Line 146-150 rewrite the paragraph by adding the DNA concentration and volumen data used in the standardization for all genes tested

Page 9           Line 234-236 rewrite the paragraph

                        Line 241 change the frase because 102 UFC/mL is not higher than 360                 UFC/mL

Page 10        line 282 change the word “bacteria” to the word  “strains” and include space after ….. respectively.

                        Line 282-285 explain the reason of this paragraph

line 306          change the word “devised” to “designed”

line 318          add the incubation temperatura of the mixture

Line 320        change the word “spiked” to “added”

Page 11        line 324          remove the space in the word “absorb bent”

line 327          the frase is not understood replace by the phrase “ 1ul of each antibody”

line 329          change the phrase to “ individual strips LFIA of 3.0 mm thick were obtained

line 333          complete the sentence like this: “DNA extraction was carried out using….”

line 360          change “events” por “pan-drug resistance genes…”

Reviewer 3 Report

The chosen topic is in line with the real trend in UTIs.

As a note, the "Discussion" section is a little bit too short. It could be made longer by comparing it to other recent studies.

This study can also have a paragraph about strengths and limitations.
